

# Initial Assessment of Landslide Prone Area using Soil Properties

Yanto[1], Arwan Apriyono[1], Purwanto Bekti Santoso[1], Sumiyanto[1]

[1] Civil Engineering Department, Jenderal Soedirman University, Purbalingga, 53371, Indonesia

*Correspondence to*: Yanto (yanto@unsoed.ac.id)

**Abstract.** Initial assessment of landslide prone area is important in designing landslide mitigation measures. This study, a part of our study on developing landslide spatial model, presents initial signal of landslide prone area. In here, we use soil depth to hardpan to assess landslide prone area in Western Central Java, a relatively small region where 23% of Indonesian landslide occurs. To this end, we interpolated soil depth to hardpan in a regular grid from irregularly distributed data. To do this, we employed three different methods: Inverse Distance Weighting (IDW), Ordinary Kriging (OK) and Co-Kriging (CK). For the
latter, we experimented with several potential covariates. To determine the best fitting model, several tests on model performance and its corresponding errors were done. Error measures used in this study are Mean Square Error (MSE), Root Mean Square Error (RMSE), Mean Absolute Error (MAE) and Mean Absolute Percentage Error (MAPE), while statistical measures employed are Standard Deviation, Variance, Interquartile Range (IQR), Mean Absolute Deviation and Median Absolute Deviation. The result shows that CK with covariate of slope and soil cohesion is the best fitting model and exhibits
clear pattern related to recorded landslide disaster sites. We found that 64% of landslide disaster events occur in the area having soil depth to hardpan of 5 – 10 m. Moreover, 84% of landslide occurrences happen in regions where soil depth to hardpan ranges from 5 to 15 m. Hence, we suggest that landslide prone area is an area possessing soil depth to hardpan of 5-15 m. This finding is advantageous for policy makers in planning and designing efforts for landslide mitigation.

## 1 Introduction

Indonesia has a large number of landslide disaster occurrences. In last decade, 3,924 landslide disaster events were recorded with 1,404 events (~36%) of them occurred in the Central Java province (BNPB 2018). Out of those numbers, 908 landslide events occurred in western part of the area. In this region, landslide disaster events have caused 211 casualties, nearly 20,000 people suffered and more than 7,000 house damaged. Moreover, the number of natural disaster in Indonesia shows increasing trend from 924 events in 2008 to 2,862 events in 2017 which accounts for more than threefold during the last decade (BNPB
2018). Considering the impact of global climate change, the number could be larger and the impact could be worsening (Banholzer et al. 2014). Hence, appropriate mitigation measures in particular locations are critical. Accordingly, understanding landslide prone area is helpful in designing policy for appropriate mitigation efforts.

Landslide occurs when safety factor (SF), defined as a ratio of shear strength and acting force (gravitational force, seepage pressure, etc) of soil layer disrupted (Das 1998). The higher the slope stability (SF > 1) the lower the possibility of landslide



to occur (Roslee et al. 2012). High slope stability exists when shear strength is high. However, soil shear strength unveils high spatial variability and is unsuitable to be directly measured due to difficulty of obtaining undisturbed sample (Apriyono et al. 2018). Nevertheless, the value can be estimated from its correlation with soil depth to hardpan which obtainable from Cone Penetration Test (CPT), one of soil test required in civil works. In this study, we define soil depth to hardpan as a depth of soil

layer capable of bearing a minimum load of 200 kg/cm$^2$.

Soil depth to hardpan is abundant as it is prerequisite for any civil structural planning and design. However, the data is usually scattered over a region. To estimate landslide prone area using soil depth to hardpan, understanding its spatial pattern is essential. To do this, applicable spatial interpolation is required, which is central contribution of this study.

Various interpolation techniques commonly used in many literatures are: statistical method such as linear regression (Lesch

and Corwin 2008; Tabari et al. 2011), geometric method such as Inverse Distance Weighting (IDW) and local polynomial (Apriyono et al. 2018; Santoso et al. 2018; Yanto et al. 2017b) and geostatistical method such as Ordinary Kriging (OK), Regression Kriging (RK) and Co-Kriging (CK) (Baskan et al. 2009; Fritsch et al. 2011; Govaerts et al. 2010; Yao et al. 2013). Comparison of interpolation methods has been done in many literatures (Siljeg et al. 2015; Wang et al. 2014; Wong et al. 2004). Performance of each method varies and highly depends on the case, location and model parameters (Siljeg et al. 2015).

Therefore, the main challenge lies on the determination of error characteristics and estimated values by testing and comparing various interpolation methods. This study is motivated by Santoso et al (2018) and aimed to improve interpolation skills of soil depth to hardpan in Western Central Java using geostatistical technique. Moreover, we employed OK and CK method with several experiments using diverse covariates to enhance IDW performance.

The structure of this paper is as follows. First, study area and data are illustrated. The interpolation methods (IDW, OK and

CK) are then presented along with potential covariates for CK method. Characteristic of errors and model performance are described both for cross-validation and prediction in the result section with analysis and discussion. The paper concludes with summary of result analysis.

## 2 Study Area and Data

This study was performed in western part of Central Java, Indonesia, bordered by Indian Ocean in the south and Java Sea in

the north (Figure 1). The elevation ranges from 0 in the edge to 3,428 m in the middle part of region. There are several mountains in the middle dominated by lithology class of tectonic and extrusive. In the coastal zone, sedimentary lithology dominates the area.

Study area chosen in this study was merely limited by data available in our laboratory which collected during the period of 2005 – 2016 with some missing data in 2008 and 2009. Over this period, we acquired 335 soil measurement sites covering an

area of 12.975 km$^2$ over 11 regencies in Western Central Java (Figure 2a). Soil properties data comprises soil depth to hardpan, soil cohesion and soil friction angle. In some locations, soil cohesion and soil friction angle are missing such that there are 227 sites possesing the data (Figure 2b,c). To implement CK method, we used soil data from ground surface, i.e. elevation and



slope. We derived and resampled elevation and slope of land data from ASTER Digital Elevation Method (DEM). There are 1,000 elevation data and 1,132 slope data resampled from 90 x 90 m ASTER DEM resolution (Figure 2d,e).

Soil depth to hardpan was used as the main soil properties to estimate landslide prone area, while soil cohesion, soil friction angle, elevation and slope data were used as covariate for CK interpolation method. We then compared the results to find the most suitable covariate.

To assess the correlation of soil depth to hardpan and landslide disaster location, we collected data of landslide occurrences from National Board of Disaster Countermeasure (BNPB) of Indonesia. There are 108 landslide events recorded containing spatial information during the period of 2011 -2017 (Figure 2f).

## 3 Methodology

In this section, methodology employed in the study is presented. First, we describe interpolation method covering IDW, OK and CK. These are methods to estimate depth to hardpan on a regular grid from irregularly distributed observations. We then present the approach for error estimate to assess which interpolation method producing the most fitting result.

### 3.1 Inverse Distance Weighting (IDW)

In IDW model, interpolation is performed by assigning weight to interpolating values – i.e. values of variable around targeted point. The weight in IDW is an inverse of interpolating variable distance obtained by dividing 1 with distance of each interpolating variable. Hence, the farther the distance, the smaller the weight (Shepard 1968). Moreover, influence of distance is governed by the value of power of distance. The higher the power, the smaller the influence of distant variable. Computation of IDW is presented below.

$$\hat{Z} = \frac{\sum_{i=1}^{n} \frac{1}{r_i^{\alpha}} Z_i}{\sum_{i=1}^{n} \frac{1}{r_i^{\alpha}}} \, , \tag{1}$$

where $\hat{Z}$ is estimated value in targeted location, $Z_i$ is value in each interpolating location, $r_i$ is distance of interpolating and targeted location and $\alpha$ is power of distance.

### 3.2 Ordinary Kriging (OK)

In anisotropic data, IDW is no longer relevant. On the other hand, OK model has an ability to take into account the effects of anisotropy on spatial data where point data falling into cluster is assigned a weight smaller than point data located outside of cluster (Boyle 2010). This occurs as the weight is given based on a function constructed from data characteristics (Isaaks and Srivastava 1989). The OK model is formulated as:

$$Z^*(u) - m(u) = \sum_{\alpha=1}^{n(u)} \lambda_{\alpha} [Z(u_{\alpha}) - m(u_{\alpha})] \, , \tag{2}$$


where u and $u_\alpha$ are vector of target location and surrounding point data given index of $\alpha$, $n(u)$ is number of point data around target location used for interpolation whereby in this study we use five point data, $m(u)$ and $m(u_\alpha)$ are expected values (mean) of $Z(u)$ and $Z(u_\alpha)$, and $\lambda_\alpha(u)$ is weight for each datum $Z(u)$ of interpolation in target location u.

The objective of OK model is to find a weight of $\lambda_\alpha$ that minimizes variance $(\sigma)$ of estimator:

$$\sigma_E^2(u) = var\{Z^*(u) - Z(u)\}, \tag{3}$$

with constraint of $E\{Z^*(u) - Z(u)\} = 0$

### 3.3 Ordinary Kriging (OK)

When data scatters with low density, both IDW and OK usually produce less realistic spatial estimates. To handle this issue,
CK technique uses spatial correlation of main variable (soil depth to hardpan) with other variables as covariate to improve the result (Myers 1984). In this study we used elevation, slope of land, soil cohesion and soil friction angle data as potential covariate. Mathematically, this approach is formulated below.

$$Z_1^*(u) - m_1(u) = \sum_{\alpha=1}^{n(u)} \lambda_{\alpha 1}[Z_1(u_{\alpha 1}) - m_1(u_{\alpha 1})] + \sum_{\alpha=1}^{n(u)} \lambda_{\alpha 2}[Z_2(u_{\alpha 2}) - m_2(u_{\alpha 2})], \tag{4}$$

where index 1 and 2 refer to main variable and covariate respectively.

### 3.4 Error Estimate

Error estimate was used to evaluate how interpolation differs from observation. This is useful for selecting the most appropriate interpolation method and interpreting the results. In this stage, we filtered covariates in CK model to yield the most suitable covariate for soil depth to hardpan interpolation.

We first conducted cross-validation approach – i.e. leave one point out and estimate the value at the leaving point. For each
interpolation method, we computed a number of error measures and compared among the methods. Table 1 summaries characteristics of error measures we used in cross-validation.

### 3.5 Interpolation Performance

Interpolation performance was evaluated by comparing gridded values and nearest observations. As the number of estimation and observation is different, we opted to assess spatial distribution of both estimation and observation for evaluation purpose.
The most suitable interpolation method is decided based on the similarity of its distribution with observation. To do this, a number of statistical measures were employed and summarized in Table 2.



## 4 Result and Discussion

In this section, we present the result and analysis in the following order: cross-validation along with analysis of error estimate, model performance with statistical measures and relationship of soil depth to hardpan and landslide disaster occurrences.

### 4.1 Cross Validation

Figure 3 shows scatterplot of estimation generated by cross-validation procedure and observation. We show a 1:1 line (red dashed line) to detect how far estimated values spread from its expected values. The closer the scattered points to this line the better the model (Yanto et al. 2017a,b). In addition, we also present a regression line (continuous blue line) to show the best fitting straight line of scatterplot along with its Pearson's coefficient of correlation ($\rho$). The slope of regression line approximating 1 indicates good representation of the model and vice versa. Moreover, the higher the $\rho$ value (maximum of 1),

the stronger the correlation of estimation and observation.

The result shows that the data scatter over the 1:1 line and regression line where larger values undergo higher spread. It is difficult to visually inspect the best fitting model from the scatter plot. However, the value of $\rho$ shows that the strongest relationship of estimation and observation occurs in CK model with slope and soil cohesion as covariate (Figure 2h). Moreover, we computed slope of regression line for each interpolation approach. The regression line slope is 0.2072, 0.3254, 0.3301,

0.3247, 0.3264, 0.3255, 0.3247, 0.3266 and 0.3491 for IDW, OK, CK-elevation, CK-slope, CK-cohesion, CK-friction angle, CK-slope-friction angle, CK-slope-cohesion and CK-all covariates respectively. Consistent with the scatterplot, quantitative measures of relationship between estimation and observation show slight difference. However, it can be inferred that CK model with covariate of slope and soil cohesion exhibits the highest Pearson's correlation coefficient, while the highest regression slope value is produced by CK-all variates.

To provide deeper insight on the comparison among interpolation approaches, we calculated quantitative error measures of MSE, RMSE, MAE and MAPE as shown in Table 3. These are error measures commonly used in many literatures (Buchwalder et al. 2006; Chai and Draxler 2014; Willmott and Matsuura 2005; Yao et al. 2013). For all error measures, the lower the value the better the interpolation performance. In Table 3, the lowest value is denoted with bold marker. As shown in Table 3, CK model with covariate of slope and soil cohesion demonstrates the most fitting model based on the value of MSE, RMSE and

MAE, while CK model with covariate of soil friction angle beats other approaches based on the MAPE value. Accordingly, we suggest that CK-slope- soil cohesion is the most relevant model for soil depth to hardpan interpolation.

### 4.2 Interpolation Performance

In cross-validation analysis, we showed that slope and soil cohesion is the best covariate for CK interpolation technique. Therefore, in this section we present analysis of model performance for IDW, OK and CK-slope-soil cohesion. For the rest of

this document, the latter will be written as CK for simplicity.





We interpolated soil depth to hardpan on 20,000 points across the study area. The result is presented in Figure 4a,b,c for IDW, OK and CK respectively along with its standard error in Figure 4d,e,f. In here, we computed standard error as the difference of interpolated values and observation mean. As shown in Figure 4a,b,c, the middle part of the study area is dominated by low soil depth to hardpan ranging from 0 to 12 m, while high soil depth to hardpan distributed in the edge of study area. Low soil

depth to hardpan also occurs in the southwestern and south-eastern part of the region. Topographically, the middle area, south-eastern and southwestern area are mountainous ranges dominated by extrusive and tectonic lithology, while the south and north side are coastal zone covered by sedimented soil (Figure 1). Hence, it can be inferred that the interpolation result is consistent with the topography and lithology of study area. Moreover, spatial spreading of standard error is quite similar to spatial distribution of soil depth to hardpan (Figure 4d,e,f). In the region where soil depth to hardpan is low, the standard error exhibits

negative values, and vice versa.

Spatial variability of soil depth to hardpan and standard error reveals that both OK and CK approaches produce quite similar pattern. On the other hand, IDW displays less spatial variability compared to OK and CK, particularly in the middle part of the region. This indicates the influence of data points situated farther from targeted point.

To obtain better understanding on the performance of IDW, OK and CK in interpolating soil depth to hardpan, we show the

distribution of observation and estimation from those three methods and shown in boxplots (Figure 5). The best model is evaluated visually from its similarity (box and whisker) with observation. As can be seen, IDW reveals the most dissimilar features to observation as it has the smallest box, the shortest whisker and the largest range of outliers. On the other hand, both OK and CK are relatively close to observation. However, it is challenging to resolve the most suitable model due to their graphical likeness.

We performed analysis on the distribution of standard error shown in Figure 6 to complement interpretation on the model performance. It can be observed that all methods generate normally distributed standard error with different shape of histogram where IDW standard error concentrates around the mean and both OK and CK disperse over the range of standard error. This implies that both OK and CK standard error has higher variability than IDW standard error. Again, it is tough to choose the best model based on visual interpretation. To resolve this, we calculated the fraction of negative and positive standard error. It

is found that 92% of IDW standard error falls below zero, indicating that IDW yields underestimate interpolation against to observational mean. On the other hand, the fraction of negative values is 63% and 64% for OK and CK respectively. In addition, the mean of standard error is -2.85, -1.07, -1.04 for IDW, OK and CK respectively suggesting that both OK and CK better represents the observation with CK overcomes OK to some extent.

To strengthen our visual inspection on the distribution of estimated values, we computed quantitative statistical measures of

soil depth to hardpan interpolated values. In here, we use common statistical measures in many literatures such as standard deviation, variance, interquartile range (IQR), mean absolute deviation (MAD) and median absolute deviation (MedAD) as shown in Table 4 (JCGM 2008; Rousseeuw and Croux 1993; Stadtler et al. 2014, Xiao-ming et al. 2016; Zwillinger and Kokoska 2000). We assess the performance of each interpolation method based on its proximity with each corresponding





statistical measure of observation. In Table 4, the most faithful values to observation are shown in bold marker. It can be seen that for all statistical measures, CK conquers IDW and OK except for MAD where IDW beats others.

Based on visual inspection and quantitative appraisal both for interpolated values and standard errors, it is suggested that CK (with soil cohesion as covariate) performs better than IDW and OK in estimating spatial distribution of soil depth to hardpan
in Western Central Java. This is consistent with the aim of Cokriging model development (Myers 1984) and its implementation (Minnitt and Deutsch 2014; Adhikary et al. 2017; Xie et al. 2018).

### 4.3 Spatial Correlation of Soil Depth to Hardpan and Landslide Occurrences

In the previous section, it was clearly shown that CK approach using soil cohesion as covariate produces the most fitting model. Hence, it is used in this section to assess spatial correlation of soil depth to hardpan and landslide occurrences. First,
we show pictorial representation of the relationship as shown in Figure 7. It can be seen that landslide disaster occurs mostly in the middle part stretching from the west to the east and in the south-eastern part of the region. Moreover, the figure displays clear spatial pattern of landslide sites related to spatial variability of soil depth to hardpan. Visually, soil depth to hardpan where landslide occurs ranges from 4 to 16 m as can be seen from the figure that in the area where soil depth to hardpan less than 4 m, there is no landslide record detected. The same feature can be examined for soil depth to hardpan being greater than
16 m.

To verify aforementioned visual assessment, we extracted the values of soil depth to hardpan in the landslide sites (or nearest location). We then grouped the value in four ranges: 0-5 m, 5-10 m, 10-15 m and 15-20 m as presented in Figure 8. In the figure, green solid square, blue solid dots, red solid triangle and black solid dots represent the number of landslide disaster events for the range of soil depth to hardpan of 0-5 m, 5-10 m, 10-15 m and 15-20 m respectively. From this number, it can be
computed that 64% of landslide occurs in the region with soil depth to hardpan ranges 5-10 m. Moreover, 84% of landslide disaster events take place in the area where soil depth to hardpan ranges 5-15 m. Hence, we advise that landslide prone area is an area holding soil depth to hardpan from 5 to 15 m.

This finding is consistent with landslide mechanism commonly occurs in Indonesia: translational and rotational landslide (Cepeda et al. 2010). In this type of landslide, it requires acting force and surface of rupture to make landslide happens. Acting
force presents in the form of gravitational force from soil mass and soil water content. Surface to rupture needs more stable underlying material (Jesus et al. 2017; Muller and Martel 2000). This can be either hard rock or soil hardpan. Hence, the depth to soil hardpan can be viewed as potential location of surface to rupture. When surface to rupture is too shallow, acting gravitational force is smaller due to low volume of soil mass and soil water content. This condition lessens the probability of landslide to occur. On the other hand, deep surface to rupture is associated with sedimentary lithology class and low slope of
land. As a result, landslide is unlikely to happen in the area with very deep surface to rupture.



## 5 Conclusion

We employed three interpolation methods: IDW, OK and CK to simulate spatial distribution of soil depth to hardpan useful for initially assessing landslide prone area in Western Central Java, Indonesia. To select the best fitting model, we performed a number of tests on the interpolated values and its corresponding error. The tests conducted include visual inspection and
quantitative measurement. In all cases, we found that visual assessment shows comparable features between OK and CK whereby they always overcome IDW. However, quantitative measures reveal that CK is better than OK in some extent. Based on this, we then compared soil depth to hardpan resulted from CK interpolation procedure and landslide disaster sites. It is clearly shown that landslide is likely to occur in the area where soil depth to hardpan ranges from 5 to 15 m. Hence, it can be said that landslide prone area is a region having soil depth to hardpan of 5-15 m. This finding is useful for policy makers in
designing mitigation efforts to landslide prone areas and subsequently to save more lives.

## 6 Acknowledgement

We thank to the Soil Mechanics Laboratory of Jenderal Soedirman University for substantial data of soil properties. In addition, we also thank to National Board for Disaster Countermeasure of Indonesia for landslide events data.

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

**Table 1. Measurements to inspect model errors in cross-validation**

| Error Measures | Formula | Reference |
|---|---|---|
| *Mean Square Error (MSE)* | $MSE = \frac{1}{n}\sum_{i=1}^{n}(p_i - o_i)^2$ | Buchwalder et al (2006) |



| Root Mean Square Error (RMSE) | $RMSE = \sqrt{\dfrac{1}{n}\sum_{i=1}^{n}(p_i - o_i)^2}$ | Willmott and Matsuura (2005); Chai and Draxler (2014) |
|---|---|---|
| Mean Absolute Error (MAE) | $MAE = \dfrac{1}{n}\sum_{i=1}^{n}|p_i - o_i|$ | Willmott and Matsuura (2005); Chai and Draxler (2014) |
| Mean Absolute Percentage Error (MAPE) | $MAPE = \dfrac{1}{n}\sum_{i=1}^{n}\left|\dfrac{(p_i - o_i)}{o_i}\right|$ | Yao et al. (2013) |

**Table 2. Statistical measures to evaluate model performance**

| Statistical Measures | Formula | Reference |
|---|---|---|
| Standard Deviation | $\sigma = \sqrt{\dfrac{1}{n-1}\sum_{i=1}^{n}(x_i - \bar{x})^2}$ | JCGM (2008) |
| Variance | $\sigma^2 = \dfrac{1}{n}\sum_{i=1}^{n}(x_i - \bar{x})^2$ | Xiao-ming et al. (2016) |
| Inter Quartile Range (IQR) | $IQR = \left(\dfrac{3\,(n+1)}{4}\right)^{th} - \left(\dfrac{(n+1)}{4}\right)^{th}$ | Zwillinger and Kokoska (2000) |
| Mean Absolute Deviation | $MAD = \dfrac{1}{n}\sum_{i=1}^{n}|x_i - \bar{x}|$ | Stadtler et al. (2014) |
| Median Absolute Deviation | $MedAD = median(x_i - \bar{x})$ | Rousseeuw and Croux (1993) |

**Table 3. Error measures of cross-validation using different interpolation methods**

| Error Measures | IDW | OK | CK | | | | | | |
|---|---|---|---|---|---|---|---|---|---|
| | | | Elevation | Slope | Cohesion | Friction Angle | Slope-Friction Angle | Slope-Cohesion | All Covariates |
| MSE | 27.2298 | 24.6058 | 31.9023 | 24.8209 | 24.5699 | 24.6738 | 24.8292 | **24.5579** | 29.1962 |
| RMSE | 5.2182 | 4.9604 | 5.6482 | 4.9821 | 4.9568 | 4.9673 | 4.9829 | **4.9556** | 5.4033 |
| MAE | 3.9436 | 3.7307 | 4.3306 | 3.7548 | 3.7284 | 3.7397 | 3.7559 | **3.7267** | 4.1821 |
| MAPE | 0.8167 | 0.8017 | 0.9119 | 0.8045 | 0.8023 | **0.8013** | 0.8053 | 0.8016 | 0.8906 |




5 **Table 4. Quantitative statistical measures of interpolation using IDW, OK and CK**

| Statistical Measures | Observation | IDW | OK | CK |
|---|---|---|---|---|
| *Standard Deviation* | 5.48 | 2.17 | 3.80 | **3.81** |
| *Variance* | 30.01 | 4.69 | 14.44 | **14.49** |
| *IQR* | 7.60 | 2.36 | 4.68 | **4.76** |
| *Mean Absolute Deviation (MAD)* | $3.01 \times 10^{-16}$ | $\mathbf{1.01 \times 10^{-16}}$ | $7.17 \times 10^{-16}$ | $7.05 \times 10^{-16}$ |
| *Median Absolute Deviation (MedAD)* | 5.04 | 1.77 | 3.50 | **3.53** |

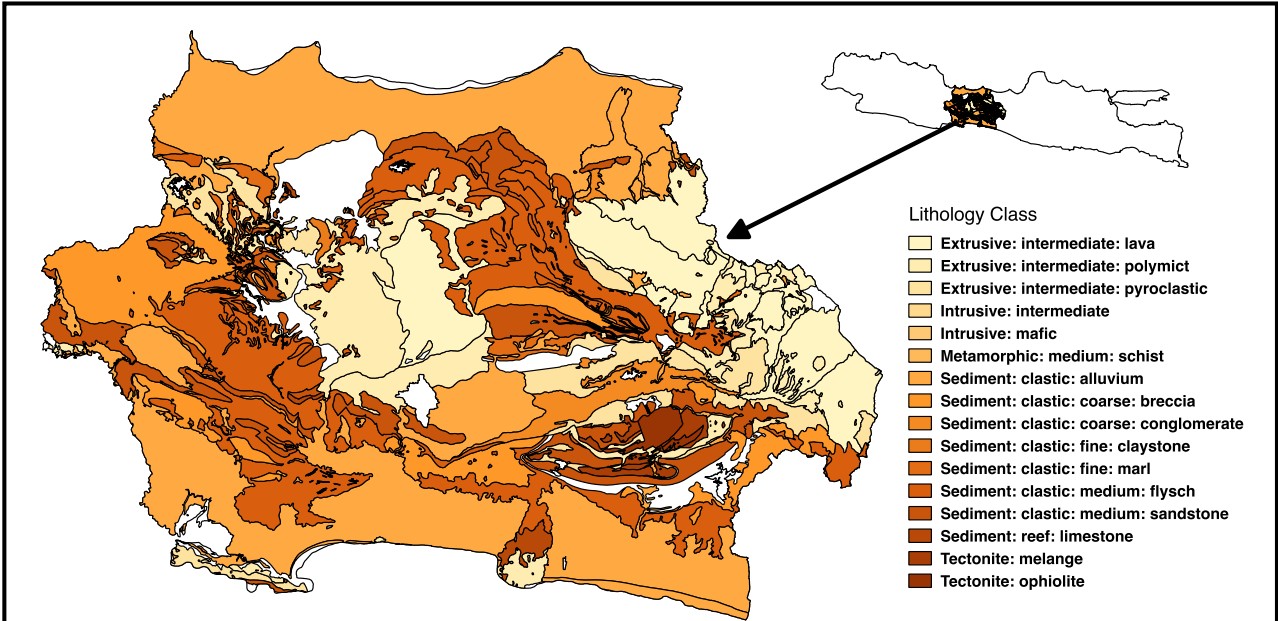

**Figure 1: Study area: Western part of Central Java along with its soil lithology.**

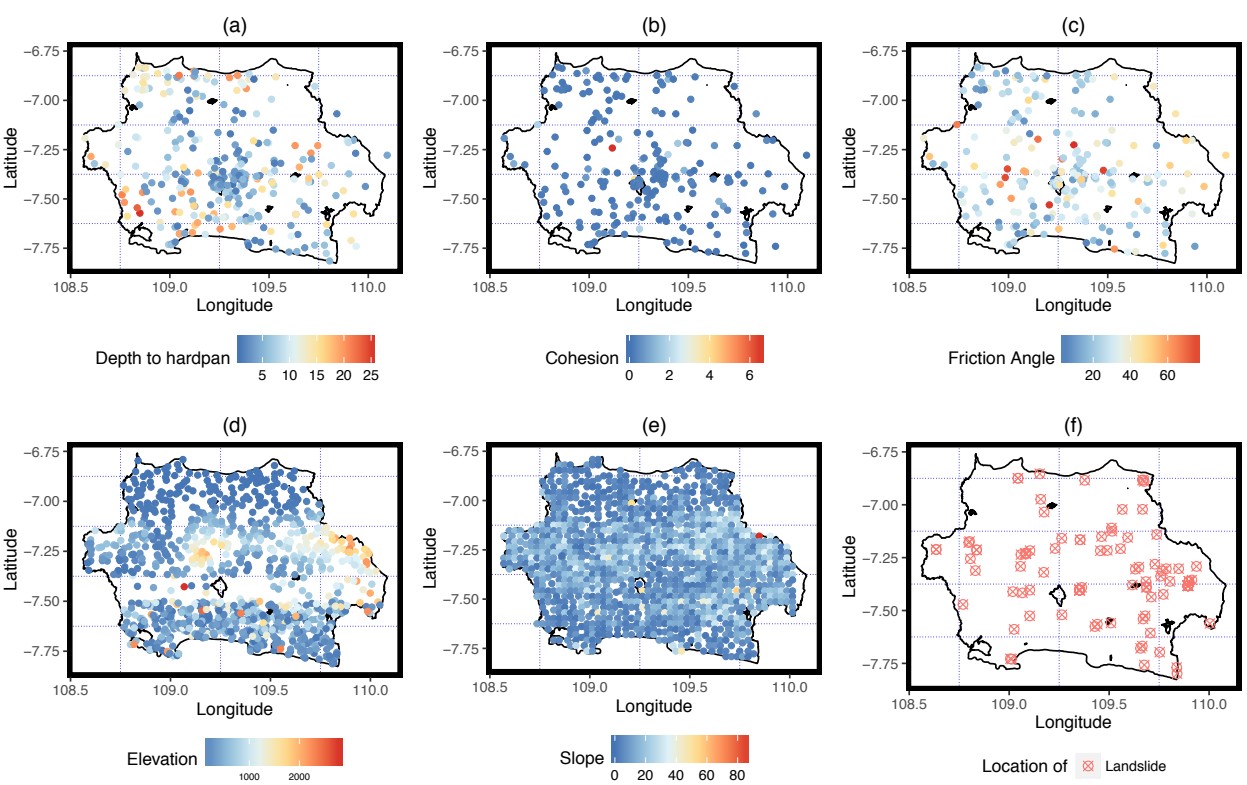

**Figure 2: Spatial variability of data: (a) soil depth to hardpan in meter, (b) soil cohesion in kN/m², (c) soil friction angle in degree, (d) ground elevation in meter above sea level, (e) slope of the land in degree, (f) location of landslide disaster events.**


**Figure 3: Scatter plot of observation and prediction from cross-validation for: (a) IDW, (b) OK, (c) CK-elevation, (d) CK-slope, (e) CK-cohesion, (f) CK-soil friction angle, (g) CK-slope-soil friction angle, (h) CK-slope-cohesion, (i) CK-all covariates.**





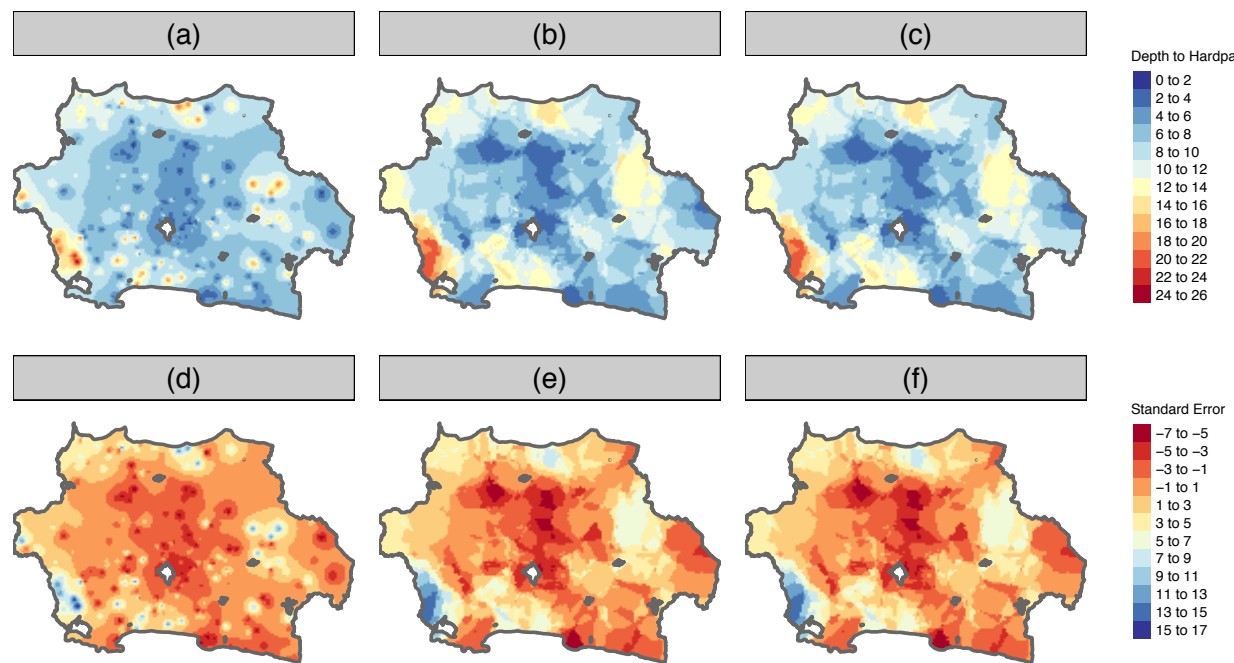

**Figure 4: Spatial distribution of estimation of soil depth to hardpan (m) for: (a) IDW, (b) OK, (c) CK and standard error (m) for: (d) IDW, (e) OK, (f) CK.**

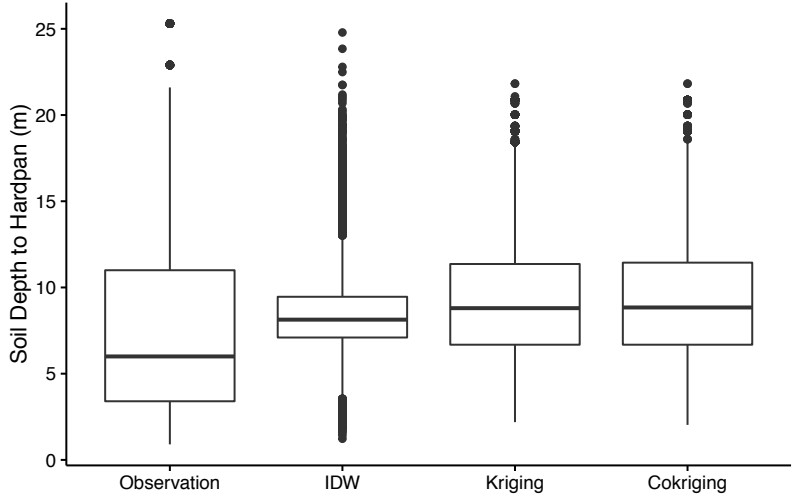

**Figure 5: Boxplot of interpolated and observed soil depth to hardpan.**

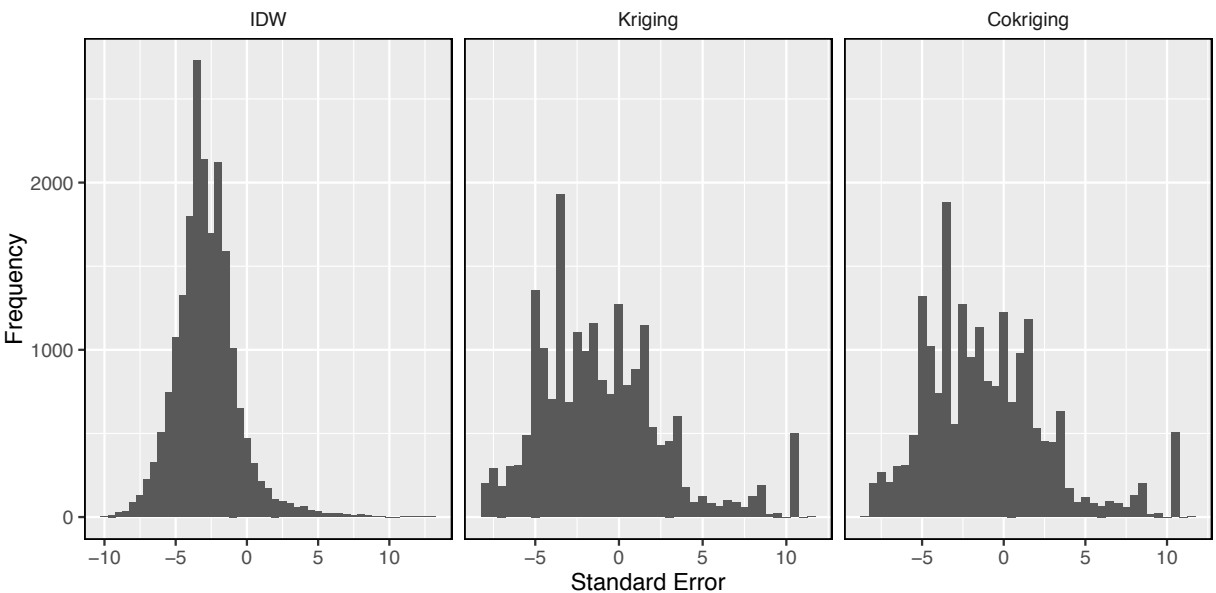

**Figure 6: Histogram of standard error (interpolated values substracted by observational mean) of IDW, OK and CK.**

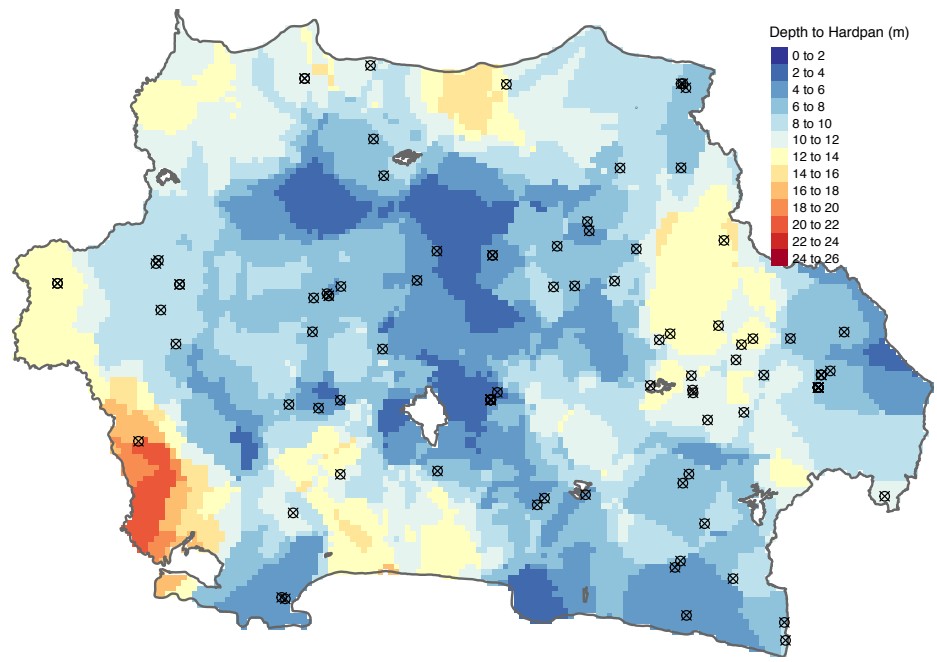

5   **Figure 7: Map of soil depth to hardpan and landslide occurrence sites.**





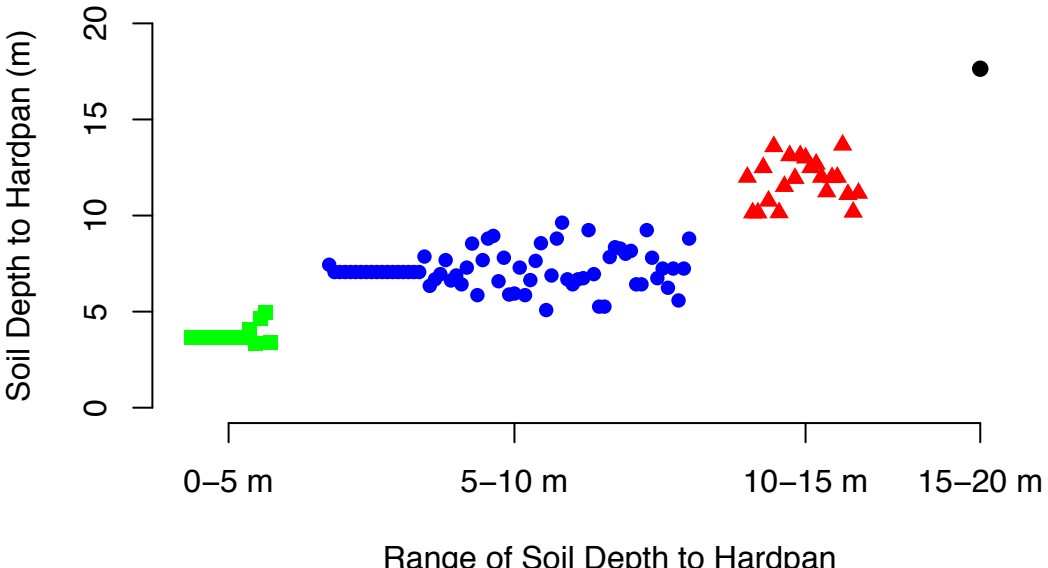

Figure 8: Map of soil depth to hardpan and landslide occurrence sites.