# Peer review of "Initial Assessment of Landslide Prone Area using Soil Properties"

_Natural Hazards and Earth System Sciences, 2019_

## Referee Comment (RC1) · Anonymous Referee #1 · 27 Jun 2019

This paper addresses the problem of soil depth distribution in the context of landslide assessments. The consideration of the spatial variability of soil depth is crucial, especially as input parameter for modeling, and therefore this paper contributes to an important topic. Different interpolation methods were tested and thoroughly validated. The results lead to a better understanding of which method is the best fit for this study, therefore it would have been great if the study area would also be reflected in the title. The paper is interesting, although certain aspects should be improved, especially in order to better understand where the focus of the study lies. The authors mention the use of covariates for Co-Kriging, a detailed information about why they did not use it for all interpolation methods or on what basis they did chose CK, would be very helpful and contribute to a better understanding. The authors state that applicable spatial interpolation is the central contribution of the study (page 2, line 8), therefore this should be mentioned in the title. In addition, another option could be to mention the study area in the title (as mentioned above), as the paper is more focused on the case study (page 2, line 17) rather than the development of new methods. It would have been interesting to discuss, if these findings are applicable to other study areas. A detailed description of the data and a characterization of the data properties used in this study would be very helpful and would increase the comprehensibility. The methods are clearly explained and if you would specify how these methods were applied in this study, for example, how many points were used for the error estimation, which parameters were used, etc., this would be a helpful addition. In the text 'landslide disaster events/occurrences' is used, but do you really mean disastrous or could 'landslide events/occurrences' also be enough here? Maybe not all events were disastrous? Is a detailed description of the structure of the paper necessary (Page 2, chapter 2), as the structure is self-explanatory? The same applies for chapter 4 and maybe it would improve readability if you could briefly explain the purpose of the following subchapters instead. Is subchapter 4.1 the corresponding result of subchapter 3.4? If this is the case, maybe you could give them the same names, as you did with 3.5 and 4.2. In chapter 4.3 there is a visual evaluation of figure 7, maybe an additional simple statistical analysis would make the results more reliable. In chapter 2, please clarify what exactly is meant by 'tectonic and extrusive lithology'? In some places, the English should be checked and corrected. Especially plural should be used more. For example - Page 1, line 7, plural on 'landslides' and occur without 's'. Page 1, line 9: exchange 'employed' by 'applied', same at line 14. Page 1, line 20: 'the' before 'last'; line 21: with 'the occurrence of' 1404 events in the Central Java province. Page 1, line 23: 'houses were damaged'. Page 1, sentence starting in line 28 is unclear. Page 2, 'is' before 'obtainable'. Page 2, line 8: 'the' before 'central'. Page 7, 'rupture surface' instead of 'surface to rupture'. Page 7, line 23, 'mechanisms' commonly 'occurring' in Indonesia. Page 8, line 8: 'landslides are' likely. Figure 3, a legend or explanation in the figure description would improve the understanding of the figures. Figure 7, legend is incomplete (landslide sites missing).

Figure 8, the x axis should have the same distribution as the y axis, also a legend would be helpful and therefore you would not have to repeat the classes in the text (page 7, line 19).

———————————————————

---

## Referee Comment (RC2) · Anonymous Referee #2 · 2 Jul 2019

The manuscript presents an application of different interpolation method (IDW, OK, CK) for obtaining the spatial distribution of soil depth to hardpan, which was then used for initial assessment of landslide prone area based on the frequency of soil depth of landslides. The authors compared and discussed the performance of different interpolation methods, followed by very limited analysis about the relationship between the soil depth and landslides. This is seriously inconsistent with the title of the manuscript. And the comparison results do not bring any new knowledge, just a simple GIS experiments. Landslide data is very important for this manuscript, but the author does not present enough information at all (only two sentences). What types of landslides in this work (soil thickness is more related to shallow landslides)? Earthquake-triggered or rainfall-triggered? Basically, all this important information is unknown. For these

reasons the manuscript cannot be accept in the present form.

---

## Author Comment (AC1) · 29 Aug 2019

We thank for the comments from Referee #2. The referee provided a very good point. Based on the referee's comments, we will add 1-2 paragraphs on the study area and data section to explain characteristic of landslides occurred in the study area. We have collected data on the description of landslide occurrences in the study area that can be found via the following link: http://geospasial.bnpb.go.id/pantauanbencana/data/datalongsorcetakall.php We found that most of landslides occurred in the study area are rainfall-induced landslide. Hence, it is relevant use soil thickness as initial indicator to assess landslide prone area. We will present a table containing detail information of the landslide events and 2 pictures of landslide events.

---

## Author Comment (AC2) · 29 Aug 2019

We thank for the valuable inputs from Referee #1. This will definitely improve our paper. The followings are our response. • Among three interpolation methods, Co-Kriging is the only method requiring co-variate. This method is basically proposed to improve the interpolation results as this method can capture any related information from co-variates related to the variate, which is not the case of two other methods, IDW and Kriging • We will add the interpolation method in the title as well as the study area • Applicability of this findings – i.e. what type of the best interpolation method – is highly dependent on the data characteristics. However, we believe that the use of soil thickness/depth as initial indicator to assess landslide prone area is applicable in other study areas • We will add the method to acquire the soil properties data. In addition,

we will also add more information on landslide data. We will add a table containing information of time, location and the cause of landslide in the study area and 2 pictures of landslide events in the study area • As we performed cross validation, therefore all data points are used for error estimation • We used the term of events/occurrences as not all these events is disastrous • We think that the description of paper structure is necessary to provide brief overview on what we present in the paper • Yes, subchapter 4.1 corresponds with subchapter 3.4. We will change the subtitle accordingly • We will make additional simple statistical analysis as suggested • We will add sentences to clarify "tectonic and extrusive lithology" • We will check the grammar carefully